# Boosting the Intra-Articular Efficacy of Low Dose Corticosteroid through a Biopolymeric Matrix: An In Vivo Model of Osteoarthritis

**DOI:** 10.3390/cells9071571

**Published:** 2020-06-28

**Authors:** Matilde Tschon, Francesca Salamanna, Lucia Martini, Gianluca Giavaresi, Luca Lorenzini, Laura Calzà, Milena Fini

**Affiliations:** 1IRCCS Istituto Ortopedico Rizzoli, Laboratory of Preclinical and Surgical Studies, via di Barbiano 1/10, 40136 Bologna, Italy; matilde.tschon@ior.it (M.T.); lucia.martini@ior.it (L.M.); gianluca.giavaresi@ior.it (G.G.); milena.fini@ior.it (M.F.); 2Department of Veterinary Medical Sciences, University of Bologna, 40064 Ozzano Emilia (BO), Italy; luca.lorenzini8@unibo.it; 3FaBit and CIRI-SDV, University of Bologna, 40136 Bologna, Italy; laura.calza@unibo.it

**Keywords:** hyaluronic acid, chitosan, corticosteroid, efficacy study, knee osteoarthritis

## Abstract

The purpose of this study was to verify the efficacy of a single intra-articular (i.a.) injection of a hyaluronic acid-chitlac (HY-CTL) enriched with two low dosages of triamcinolone acetonide (TA, 2.0 mg/mL and 4.5 mg/mL), in comparison with HY-CTL alone, with a clinical control (TA 40 mg/mL) and with saline solution (NaCl) in an in vivo osteoarthritis (OA) model. Seven days after chemical induction of OA, 80 Sprague Dawley male rats were grouped into five arms (n = 16) and received a single i.a. injection of: 40 mg/mL TA, HY-CTL alone, HY-CTL with 2.0 mg/mL TA (RV2), HY-CTL with 4.5 mg/mL TA (RV4.5) and 0.9% NaCl. Pain sensitivity and Catwalk were performed at baseline and at 7, 14 and 21 days after the i.a. treatments. The histopathology of the joint, meniscus and synovial reaction, type II collagen expression and aggrecan expression were assessed 21 days after treatments. RV4.5 improved the local pain sensitivity in comparison with TA and NaCl. RV4.5 and TA exerted similar beneficial effects in all gait parameters. Histopathological analyses, measured by Osteoarthritis Research Society International (OARSI) and Kumar scores and by immunohistochemistry, evidenced that RV4.5 and TA reduced OA features in the same manner and showed a stronger type II collagen and aggrecan expression; both treatments reduced synovitis, as measured by Krenn score and, at the meniscus level, RV4.5 improved degenerative signs as evaluated by Pauli score. TA or RV4.5 treatments limited the local articular cartilage deterioration in knee OA with an improvement of the physical structure of articular cartilage, gait parameters, the sensitivity to local pain and a reduction of the synovial inflammation.

## 1. Introduction

Osteoarthritis (OA) is the most prevalent form of joint disease and affects all joint structures via multiple factitive pathways [1]. OA particularly affects middle-aged and elderly people (approximately one out of three people over the age of 65). About 250 million people worldwide are symptomatic OA patients, of whom 15 million are in the European Union [2]. By 2050, according to United Nations projections, the number of people affected by OA is expected to increase due to population ageing and increased life expectancy [3]; about 130 million people will suffer from OA worldwide, of whom 40 million will be severely disabled, resulting in a significant burden at the society level. In addition, due to asymptomatic patients, the real burden of OA is unmeasurable [4].

OA is considered a whole joint disease as cartilage and subchondral bone are subject to irregular external mechanical stress or morphological alterations, which result in loss of function in absorbing biomechanical forces [5]. Likewise, muscles, ligaments, and menisci undergo alterations owing to injury or weakness, leading to a breakdown in function and amplification of physical stresses. Finally, synovium and related synovitis are responsible for the inflammatory responses that lead to further damage to the surrounding tissues due to the release of pro-inflammatory mediators and have also been implicated in the initiation of cartilage loss in healthy joints [6].

OA pain is the dominant and most disabling symptom to fight and to face by clinicians; it is the major driver for the choice of treatment to be adopted [6]. In fact, functional limitations, joint instability, depression and loss of independence are all consequences of OA pain that, in turn, reduces quality of life. OA management involves pharmacological and non-pharmacological options, but when these fail to relieve symptoms, joint replacement surgery is considered. Non-pharmacological approaches are often underutilized. Considering the pharmacological approaches, they include systemic and/or local treatments based on steroidal, nonsteroidal anti-inflammatory drugs (NSAIDs), analgesics and hyaluronic acid (HY) infiltrations [7]. To date, oral administration of NSAIDs and analgesics is the most common treatment for OA-related symptoms [8]. However, the use of these therapies is restricted to short periods of time and at lowest possible doses due to side effects [9,10]. The risk of the onset of these effects can be bypassed by intra-articular (i.a.) administration. The main disadvantage of this administration is that the effects tend to be short-lived at the joint level [11]. In addition to site-specific administration of NSAIDs and analgesics, HY and corticosteroids are commonly used intra-articularly. Infiltrations of HY have anti-inflammatory, viscosupplementation and chondro-protective effects and stimulate the synthesis of proteoglycans and extracellular matrix [12]. However, due to the short duration of action, more infiltrations are needed [13]. Intra-articular infiltrations of corticosteroids can reduce painful symptoms up to three weeks after the infiltration [14], but they can alter and/or inhibit the regeneration of articular cartilage [15,16,17].

Recently, an in vivo study conducted by our group tested the infiltrative chondroprotective effects of HY associated to the anabolic properties of chitlac matrix (CTL) for knee OA. The study demonstrated that the combined administration of HY-CTL limits the development of lesions at the level of the articular cartilage, restoring the normal physiologic microenvironment and also limiting the alterations of the neighboring structures (i.e., the synovial membrane) [18]. Additionally, a reduction in galectin-1 and galectin-2 expression was seen in animals treated with HY-CTL [18]. Galectins seem to be highly expressed and secreted by inflamed synovium in OA patients; in particular, galectin-1 and -3 over-expression was observed in OA cartilage [19,20]. CTL, a derivative of chitosan, is a biocompatible modified polysaccharide obtained by reductive amination of the primary amines of the polymer with lactose. Previous studies have shown that CTL induces cell aggregation when cultured with primary pig chondrocytes, underlining its intrinsic biological and anabolic activity [21]. More recently, the effects of triamcinolone acetonide (TA) delivered by HY-CTL matrix in inflamed primary human articular chondrocytes were tested, showing that their association increased cell viability and reduced pro-inflammatory cytokine secretion, thus limiting the cytotoxic effects of TA alone [22].

The hypothesis of the present study was that the i.a. delivery of HY, CTL and corticosteroid, by combining pharmacological, biological and viscosupplementation activities, would have beneficial effects on OA progression and OA-related joint pain.

To this end, the purpose of the study was to evaluate the efficacy of a single intra-articular injection of HY-CTL, enriched with two different concentrations of TA, in comparison with HY-CTL alone, with a commercial and clinically accepted control (TA based), and with saline solution in an in vivo model of OA induced by monoiodoacetate (MIA) injection in the rat knee joint. In detail, effects of treatments on histopathological joint aspects, inflammation, pain sensitization and gait were evaluated.

## 2. Materials and Methods

### 2.1. Material Formulations

Kenacort^®^ (Bristol-Myers Squibb, Lot: AAU5788) is a commercially available TA drug and was purchased as a sterile vial with a concentration of 40 mg/mL. Saline (0.9% NaCl, Fresenius Kabi Italia, Lot: 19LK20WA) is a commercially available product which was purchased in sterile vials. HY-CTL is a steam sterilized (121 °C/15 min) viscous solution composed of: HA (1.25%,weight/volume (w/V)), chitlac (0.75%, w/V), hydroxypropyl-β-cyclodextrin (4.00%, w/V), sorbitol (2.80%, w/V), polysorbate 80 (0.50%, w/V), sodium chloride (0.24%, w/V), di-hydrogen potassium phosphate (0.15%, w/V) and water per injection. RV2 (lot no. RV2_13092018) and RV4.5 (lot no. RV45_13092018) are steam sterilized (121 °C/15 min) viscous solutions with the same composition of HY-CTL and TA 0.20% and 0.45% w/V, respectively.

### 2.2. Experimental Design and OA Induction

The study was performed according to European and Italian regulations on animal experimentation (2010/63/EU and Law by Decree 26/2014). The experimental protocol was approved by the Ethical Committee of the Istituto di Ricerca e Cura a Carattere Scientifico (IRCCS) Istituto Ortopedico Rizzoli and Italian Ministry of Health (authorization no. 278/2018-PR, 16 April 2018). The Animal Research: Reporting of In Vivo Experiments (ARRIVE) guidelines were followed in the manuscript preparation [23].

An a priori power analysis was carried out using the G*Power software v.3.1.9.2 (Franz Faul, Universität Kiel, Germany) to estimate the sample size of *n* = 5 different intra-articular treatments. It was hypothesized that the effect size would have been f ≥ 0.40 for histomorphometric scores (one-way ANOVA); by considering α = 0.05 and 1 − β = 0.95, a minimum number of n = 16 animals for experimental treatment was needed.

Eighty adult male Sprague Dawley rats (250 ± 0.3 g, 9 ± 1 weeks, ENVIGO RMS Srl, S. Pietro al Natisone, Udine, Italy) were stabled under controlled conditions (55 ± 5% relative humidity and 20 ± 0.5 °C) and supplied with standard diet and water ad libitum. During the experiment, animals were always housed two per cage with enriched materials.

At time zero (baseline), monosodium iodoacetate (MIA, Sigma Aldrich Merck, cat. no. I2512) was dissolved in sterile saline solution. Right knee joints were injected with single i.a. suspension of MIA (1 mg/50 µL), while contralateral joints were left untreated. Seven days after MIA administration, animals were grouped into 5 experimental groups (*n* = 16 each) and their knees were injected with 50 µL of:-Group 1: TA (Kenacort^®^ 40 mg/mL).-Group 2: HY-CTL.-Group 3: RV2 (HY-CTL + 2.0 mg/mL TA).-Group 4: RV4.5 (HY-CTL + 4.5 mg/mL TA).-Group 5: NaCl (0.9% sterile saline solution).

No post-operative analgesic or antibiotic therapies were administered, and after treatments, animals were returned to their cages with free access to food and water. Twenty-one days after treatment, animals underwent terminal procedures under general anesthesia induced by i.m. injection of ketamine (87 mg/kg) and xylazine (3 mg/kg). Animals were euthanized with an i.v. injection of Tanax (0.3 mL/kg, Hoechst AG, Frankfurt-am-Mein, Germany). The entire closed joint capsule of the right knee was dissected and removed from each animal. Any adverse observations at the injection sites were described.

### 2.3. Local Pain Sensitivity and Gait Measurements

The pain sensitivity to a mechanical stimulus applied to the right knee joint and gait analysis were measured at baseline, 7 days after MIA induction and 7, 14 and 21 days after i.a. injection of treatments. A pressure application measurement (PAM) device (Ugo Basile, Italy), consisting of a force transducer mounted on the operator’s thumb and a power recording unit, was used to measure the maximum mechanical force which elicits the a response from the animal. Briefly, rats were restrained by hand and the hind paw was lightly pinned with a finger to hold the knee in flexion at a similar angle for each rat (Figure 1). With the knee in flexion, the PAM transducer was pressed against the lateral side of the ipsi-medial knee while the operator’s thumb lightly held the lateral side of the knee. The PAM software guided the user to apply an increasing amount of force at a constant rate (300 grams per second) [24]. If the rat tried to withdraw its knee, or if it showed any behavioral signs of discomfort or distress, such as freezing of whisker movement or wriggling or vocalizing, the test endpoint was reached, and the force recorded. Three measurements at 1 min intervals were taken for each knee, and the withdrawal force data were averaged. Measures of the peak force that elicited a response were expressed as grams-force (gf).

The CatWalk apparatus (Version 10.5, Noldus Information Technology, Wageningen, Netherlands) was used to perform the gait analysis. In a dark environment, animals walked voluntarily on a glass walkway with a fluorescent light beamed into the floor. As they walked, their footprints were recorded by a camera present under the glass. The CatWalk XT software connected to the apparatus calculated the statistics related to dimensions, time, distances, and relationships between footprints, allowing us to assess the gait and locomotion function of these animals. Animals were trained for one week before the execution of the test to avoid any stress; briefly, animals were positioned in the gait apparatus in the dark and were free to move. During the test, all animals performed 3 uninterrupted compliant runs at each time-point. To stimulate animals to execute the test, their cages were placed at the end of the catwalk. Three consecutive runs per animal at each time-point were analyzed and the following parameters were considered in the analysis:-Paw area: the total floor area contacted by the paw during the stance phase.-Stand: the duration in seconds of contact of a paw with the glass plate.-Swing: the duration in seconds of no contact of a paw with the glass plate.-Single stance: the duration in seconds of contact with the glass plate with only one paw.

In order to avoid bias related to circadian oscillations, pain sensitivity and CatWalk tests were performed at each experimental time between 10–12 a.m. in the morning. Measurements of the right hind limbs were normalized by dividing them with their contralateral left hind limbs.

### 2.4. Histology

Twenty-one days after the i.a. injection of treatments, entire closed right joints were placed in 10% formalin buffer and processed for paraffin embedding procedures. Samples were decalcified in 5% solution of formic and hydrochloric acid for about 48 h. The joints were then extensively rinsed in distilled water, dehydrated in graded alcohol solutions (70%, twice in 95% and three times in 100%; 60 min for each step), cleared in xylene and finally paraffin embedded. Frontal sections (5 ± 1 μm) were taken through the entire joint by a semi-automated microtome (HM Leica microtome) and 20 sections were obtained from each joint. Three slides were stained with Safranin O/Fast Green while the other 2 slides were stained with hematoxylin and eosin (H&E). Histological images were taken with a digital pathology slide scanner (Aperio-Scanscope AT2, Leica Biosystems, Wetzlar, Germany).

The slides stained with Safranin O/Fast Green were scored using two modified semi-quantitative grading scores: the Osteoarthritis Research Society International (OARSI) and the Kumar scores, evaluated at magnifications of 10×, 20× and 40× [25,26]. OARSI and Kumar scoring were performed separately to all four quadrants (lateral femoral condyle (LFC), lateral tibial plateau (LTP), medial femoral condyle (MFC) and medial tibial plateau (MTP)). The slides stained with H&E were scored using the Krenn modified semi-quantitative grading score and evaluated at magnifications of 10×, 20× and 40× [27]. This score was performed in the medial and lateral aspects to evaluate synovial inflammation.

Finally, sections stained with Safranin O/Fast Green and H&E were used for the histological assessment of menisci (medial, MM, and lateral, LM) using the Pauli semi-quantitative score [28].

### 2.5. Immunohistochemistry

Two additional slides from each specimen were dewaxed in decreasing graded ethanol solutions, rinsed for 10 min in phosphate buffered saline (PBS) and then immunostained for matrix type II collagen and aggrecan. Briefly, after fixation, sections were rinsed in PBS and permeabilized by incubation in 0.3% hydrogen peroxide in PBS solution for 15 min. Slides were pre-treated for antigen unmasking with 0.2% Pronase (Sigma–Aldrich, St. Louis, MO, USA) solution in PBS for 30 min at 37 °C. After washing, the slides were incubated at room temperature for 1 h with Blocking Serum (Vectastain Universal-Quick-Kit, Vectors Laboratories, Burlingame, CA, USA) to prevent nonspecific binding, followed by incubation with specific rabbit polyclonal antibodies against type II collagen and aggrecan (NSJ Bioreagents, San Diego, CA, USA) overnight at 4 °C. After rinsing in PBS, the slides were incubated with an anti-rabbit Horseradish peroxidase (HRP)-conjugated secondary antibody (Bethyl Laboratories, Montgomery, TX, USA) and streptavidin/peroxidase complex (Vectastain Universal Quick-Kit). Finally, reactions were developed using the Vector NovaRed Substrate Kit for Peroxidase (Vectors Laboratories, Burlingame, CA, USA). Negative controls, by omitting the primary antibody, were included to check the proper specificity and performance of the applied reagents.

### 2.6. Statistical Analysis

Statistical analysis was performed using R v.3.6.1 software (R Development Core Team, 2014) and R packages ‘ordinal’ v.2019.4-25 [29], ‘lme4’ v. 1.1-21 [30], ‘lmerTest’ v.3.1 [31], ‘emmeans’ v.1.4.1 [32], and ‘ggplot2’ v.3.1.1 [33]. Ordinal data are presented as the median (interquartile range (IQR)), while continuous normally distributed data are presented as the mean ± standard error of the mean (SE), at a significance level of *p* < 0.05. Normal distribution (Shapiro–Wilk normality test) and homogeneity of variance (Levene test) were verified before doing data analysis.

Cumulative link models (CLM) were used to test if significant effects or interactions of the factors ‘treatment’ and ‘joint compartment’ on histological scores were present. Linear mixed models (LMM) were used to evaluate if there were significant interactions or effects of the ‘treatment’ factor (between-subjects) and the ‘experimental time’ factor (within-subjects) on PAM and CatWalk parameters. Pairwise comparisons of estimated marginal means were carried out as post-hoc tests to identify significant differences among groups in terms of effect size (*d*_msw_; hereinafter referred to as *d*) [34]. Sidak’s adjusted *p*-values were calculated.

## 3. Results

### 3.1. Clinical Observations

Two animals treated with TA died four and six days before the foreseen experimental endpoint: tissues were not considered useful for microscopic analysis due to the decomposition following lytic processes. Body weight was maintained in the first week following i.a MIA injection in all experimental groups.

### 3.2. Local Pain Sensitivity and Gait Measurements

Joint pain thresholds measured by PAM at baseline (BASAL), seven days after MIA (MIA) and 7, 14 and 21 days after i.a. treatments are reported in Figure 1A–C. MIA infiltrations induced a significant decrease (d = −2.3; *p* < 0.0005) in pain threshold in comparison with healthy joints at baseline.

Seven days after treatment, the PAM values of RV2 and RV4.5 were significantly higher than those of MIA (1.7% and 3.7%, *p* < 0.0005), higher than those of TA (2.2% and 2.7%, *p* < 0.0005) and NaCl (1.8% and 2.3%, *p* < 0.0005) and close to baseline values (BASAL).

HY-CTL, like RV2 and RV4.5, had PAM values close to those of BASAL and significantly higher than those of TA (1.6%, 2.3% and +2.1%, *p* < 0.0005) and NaCl (2.1%, 2.7% and 2.6%, *p* < 0.0005). Finally, at 21 days, a decrease of the pain threshold of the RV2 and RV4.5 groups was observed, approaching again close to MIA values; RV2 and NaCl had significantly lower values compared to HY-CTL (−1.7% and −1.6%, *p* < 0.0005). TA, HY-CTL and NaCl did not change over time from MIA values. Gait analysis measured by CatWalk at baseline (BASAL), seven days after MIA (MIA) and 7, 14 and 21 days after i.a. treatments, adjusted by rat body weight, are reported in Figure 2 and Figure 3. Significant interactions of the ‘treatment’ and ‘experimental time’ factors were observed for the Stand, Paw Area and Single Stance parameters; the Swing parameter showed significant effects of both factors.

Significant changes in all gait parameters were found between the BASAL and MIA values (*p* < 0.0005). At all experimental times, Stand values (Figure 2A,B) measured for RV4.5 did not differ from those of TA, which were not different from BASAL values. At 14 days, they were significantly higher than those of NaCl (1.4% and 2.2%, *p* < 0.05 and 0.0005, respectively). At 21 days, RV2 values were significantly lower in comparison with NaCl, HY-CTL and TA (−2.8%, −3.1% and −4.7%, *p* < 0.0005).

The Paw Area results (Figure 2C,D) showed that at all experimental times, RV4.5 and TA did not differ from BASAL values and they had significantly higher results compared to RV2 (1.6%, *p* < 0.05). At 14 and 21 days, RV4.5 values were significantly higher than those of NaCl (1.5% and 1.6%, *p* < 0.05 and *p* < 0.005, respectively). Swing values (Figure 3A) for TA and RV4.5 showed similar results: they were significantly lower than those of NaCl (d = −1.1 and d = −0.9, respectively) and different from HY-CTL (d = 1.0 and d = −0.9, respectively).

Single Stance values (Figure 3B,C) of RV4.5 were close to BASAL ones at all post-MIA experimental times. The results of Single Stance of RV4.5 and TA were significantly higher than the NaCl groups at all experimental times. Moreover, at day 7, RV4.5 values were higher than those of HY-CTL (1.6%, *p* < 0.05); at 14 and 21 days, TA was significantly higher in comparison with HY-CTL (2.0% and 1.8%, *p* < 0.05) and RV2 (5.6% and 2.8%, *p* < 0.0005), which were significantly lower than those of BASAL.

### 3.3. Histology

As shown in Figure 4A–E, right knee joints (both the lateral and medial compartments) treated with TA and RV4.5 showed greater cartilage staining with Safranin O (indicating the presence of glycosaminoglycans) and reduced fibrillation, clones and fibrous tissue in comparison to joints treated with HY-CTL and NaCl. Animals treated with HY-CTL showed reduced fibrillation, clones, and fibrous tissue in comparison to RV2 and NaCl, where inflammation extending into the articular cartilage and bone with minimal-to-moderate pannus formation were seen (Figure 4A–E). The semi-quantitative evaluations by OARSI and Kumar scoring (Figure 4F–G) conveyed significantly lower results in the medial than in the lateral compartment, and higher results in the tibial plateau than in the femoral condyle (40% and 22%, respectively). Both scores showed similar results, highlighting that all groups had significantly lower values than NaCl-treated joints. Regarding OARSI scores among treatments, TA had significantly lower values than HY-CTL (−48%, *p* < 0.005) and RV2 (−85%, *p* < 0.0005). RV4.5 had significantly lower values in comparison with RV2 (−38%, *p* < 0.0005). Regarding Kumar scores, TA had lower values than HY-CTL (−86%, *p* < 0.0005) and RV2 (−124%, *p* < 0.0005). Finally, RV4.5 had significantly lower values in comparison with HY-CTL (−31%, *p* < 0.005) and RV2 (−43%, *p* < 0.0005).

Concerning the evaluation of menisci (Figure 5), TA and RV4.5 showed the presence of slight undulation on the inner board of the menisci and a normal or diffused cellularity. In HY-CTL and RV2, a moderate fibrillation on the femoral side and on the inner board and the presence of hypocellular and/or acellular areas were seen in the lateral menisci. Areas of fibrocartilaginous separation with unorganized collagen fibers were seen in menisci treated with NaCl (Figure 5A–E). Semiquantitative Pauli score showed that RV4.5 achieved significantly lower results than NaCl (−25%, *p* < 0.05). Independently of the treatment, most of measurements corresponded to a grade 2 mild degeneration of menisci (Figure 5F).

Histological evaluation of the synovial membrane highlighted an enlargement of the synovial lining cell layer and an increased density of the cells in the synovial stroma in joints treated with HY-CTL, RV2 and NaCl in comparison with joints treated with TA and RV4.5 (Figure 6A–E). The histological aspect was confirmed by Krenn score (Figure 6F), demonstrating that all treated groups had significantly lower values than NaCl-treated joints (TA: −31%, *p* < 0.0005; HY-CTL: −18%, *p* < 0.0005; RV2: −12%, *p* < 0.005; and RV4.5:−30%, *p* < 0.0005). Among the treatments, TA had significantly lower values than HY-CTL (−18%, *p* < 0.005) and RV2 (−27%, *p* < 0.0005). Values for RV4.5 were significantly lower in comparison with HY-CTL (−14%, *p* < 0.005) and RV2 (−20%, *p* < 0.0005).

### 3.4. Immunohistochemistry

Representative images of immunohistochemical staining for type II collagen and aggrecan are shown respectively in Figure 7 and Figure 8. As observed, type II collagen and aggrecan showed more intense staining in RV4.5 and TA in comparison to joints treated with HY-CTL and RV2. NaCl-treated samples had the least intense expression of type II collagen and aggrecan in comparison to all the other treatments.

## 4. Discussion

To date, pharmacological options used for OA management remain largely palliative [2] and innovative treatments are still at the research and development stage. Most of them comprise options with intra-articular delivery (i.e.,platelet rich plasma, PRP, fibroblast growth factors FGF, gene therapy, teriparatide, Wnt specific signal inhibitors) aimed to maximize efficacy and minimize systemic toxicity [35,36].

In this study, it was hypothesized that the combination of HY, CTL and a commonly used corticosteroid (triamcinolone acetonide) would exert beneficial effects in knee OA by coupling the chondroprotective effect of HY, the anabolic properties of CTL and the anti-inflammatory action of TA.

To this aim, degenerative histopathological alterations and inflammation, induced in a preclinical model of chemically induced OA, were evaluated at the microscopic level, and OA related pain was measured by gait analyses and local pain sensitivity.

In this study, the i.a. treatment with TA (40 mg/mL) and RV4.5 (TA concentration: 4.5 mg/mL) showed a histopathological improvement of the physical structure of articular cartilage and a reduction of synovial membrane inflammation that, in turn, led to significant improvement in the gait behavior of rats. In particular, joints treated with RV4.5 and TA had significantly lower OARSI and Kumar cartilage scores than those treated with NaCl and RV2, a lower grade of synovitis than those treated with NaCl, HY-CTL and RV2, as measured by Krenn score, and a lower grade of meniscal degeneration than those treated with NaCl, as measured by Pauli score. In addition, type II collagen (the major extracellular component of articular cartilage) and aggrecan (the major proteoglycan of articular cartilage responsible for the ability of the tissue to withstand compressive loads) showed a more intense expression in TA and RV4.5 treatments. Gait parameters were analyzed repetitively during experiments and significant results were obtained from seven days after the i.a. treatments. All parameters of gait behavior for TA and RV4.5 treated rats significantly improved at each experimental time-point in comparison with other treatments and without differences from baseline healthy joints. As far as the local arthritic joint pain measurement was concerned, RV4.5 and RV2 significantly decreased the sensitivity to local pain in comparison with TA and NaCl without differences from baseline thresholds.

In our study, we adopted a chemical model of OA induction by delivering MIA in the knee of rats. From scientific reports, the MIA model is a rapidly developing model that allows us to focus on the pain and degenerative aspects of OA pathophysiology [37]. In fact, i.a. infiltration with MIA inhibits glyceraldehyde-3-phosphate dehydrogenase of the Krebs cycle, leading to chondrocyte degeneration and necrosis with the involvement of the entire articular cartilage, both at the femoral and tibial compartments. At the same time, experimental changes and degeneration were also detected at the subchondral bone, with an increase in the number of osteoclasts and osteoblasts [38]. In addition to joint degeneration and pain, the MIA model also generates acute and transient inflammation, like that observed in human OA patients [37], and clear gait disturbances related to pain responses [39]. This OA model, which avoids the need for surgery, is widely used in short term studies aimed at evaluating the efficacy of drugs for reducing joint pain [40,41]. For the above-mentioned reasons, we measured histopathological scores at both femoral condyles and tibial plateaus and separately in the medial and the femoral compartments, founding different results. In our series, the lateral compartment was more compromised than the medial one, which was probably caused by the lateral approach for MIA injections; however, our scoring data at baseline (which for OARSI reached near the worst results), were in line with data obtained by other authors adopting the same model [42].

For OA studies, Stand and Swing durations and Paw Print Area are commonly reported parameters measured by CatWalk in gait analysis [43,44,45,46,47]. For gait and pain sensitivity measurements, a biphasic response is reported in the literature, characterized by an early synovial inflammatory phase up to day 7, and a second chronic phase from day 14 corresponding to structural and degenerative alterations [48]. Thus, in our study, by performing three sequential evaluations after treatment delivery, their effects on both phases were considered. In our series, rats treated with NaCl showed shorter Stand durations, higher Swing durations and smaller Print Area, meaning that rats were unwilling to touch the CatWalk platform. They maintained OA limb retraction and avoided loading their body weight onto the treated limb when walking. Instead, treatment with TA and RV4.5, by ameliorating knee pain at each experimental time, were able to reverse these behavioral gait changes. Since body weight is a relevant parameter, it was measured at each time-point and, for CatWalk analysis, data were adjusted by their body weight. Local pain sensitivity measurements confirmed that treatments combining HY-CTL and TA were able to switch off local pain in a dose-dependent manner and even at earlier time-points, presumably thanks to the amount of TA immediately available due to cyclodextrin. PAM values at baseline and at MIA agreed with literature data, being about 800–1000 gf and 500 gf, respectively [24,49].

This work has some limitations. Firstly, MIA is a glycolysis inhibitor that causes widespread chondrocyte death and, consequently, damage to articular cartilage with fibrosis and bone remodeling. Thus, MIA pathogenesis does not closely mimic the clinical etiology of human OA, but in rodents, MIA induces antalgic gait and compensatory behaviors similar to humans and is very useful to test novel analgesic drugs [39]. Secondly, gender differences were not considered as far as the MIA model was concerned or in the efficacy of treatments, since only male rats were used. Thirdly, measurements of local inflammatory markers are lacking, even if Kumar and Kreen scores globally considered the presence of reactive and inflammatory cells.

In conclusion, this study demonstrated that the combination of HY, CTL and TA produced an improvement in knee articular cartilage degeneration, synovium inflammation and gait and pain behavior in a preclinical model of OA. Further investigations in larger animal models and the analysis of gender differences in OA development and the therapeutic efficacy of treatments should be considered.

## Figures and Tables

**Figure 1 cells-09-01571-f001:**
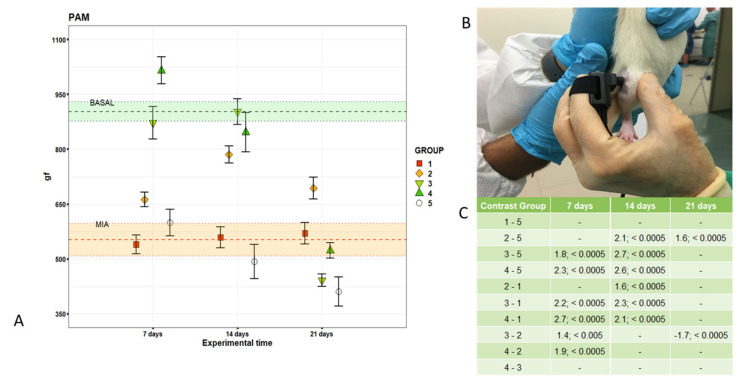
Local pain sensitivity (pressure application measurement (PAM)) results. (**A**) Dot plot of PAM for each treatment measured at treated knee joints at each experimental time; dots indicate mean and error bars indicate standard error (SE). Pain threshold values (mean and SE) measured at baseline (BASAL) and seven days after MIA (MIA) are reported as green and orange zones. The selected Linear mixed models (LMM) model showed a significant interaction of treatment and experimental time factors (F = 10.46, *p* < 0.0005). (**B**) Image of measurement of the limb withdrawal threshold. (**C**) Pairwise comparisons (*d*-value; *p*-value). Group 1: TA; group 2: HY-CTL; group 3: RV2; group 4: RV4.5; group 5: NaCl. gf: grams-force.

**Figure 2 cells-09-01571-f002:**
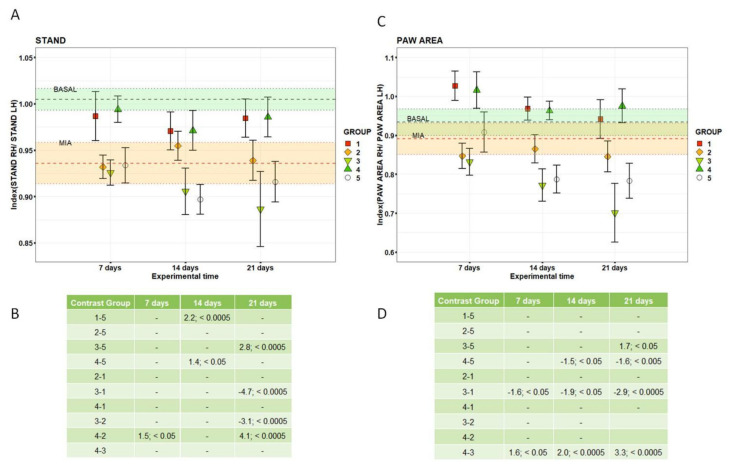
CatWalk results. Dot plots (**A**,**C**) and table reporting pairwise comparisons (*d*-value; *p*-value) (**B**,**D**) of the Stand results and Paw Area for each treatment measured at each experimental time. Dots indicate mean and error bars indicate SE. Stand and Paw Area values (mean and SE) measured at baseline (BASAL) and seven days after MIA (MIA) are reported as green and orange zones. The selected LMM models showed significant interaction of treatment and experimental time factors for Stand values (F = 5.43, *p* < 0.0005) and Paw Area (F = 3.40, *p* < 0.0005). Group 1: TA; group 2: HY-CTL; group 3: RV2; group 4: RV4.5; group 5: NaCl. Right hind paw (RH) and left hind paw (LH).

**Figure 3 cells-09-01571-f003:**
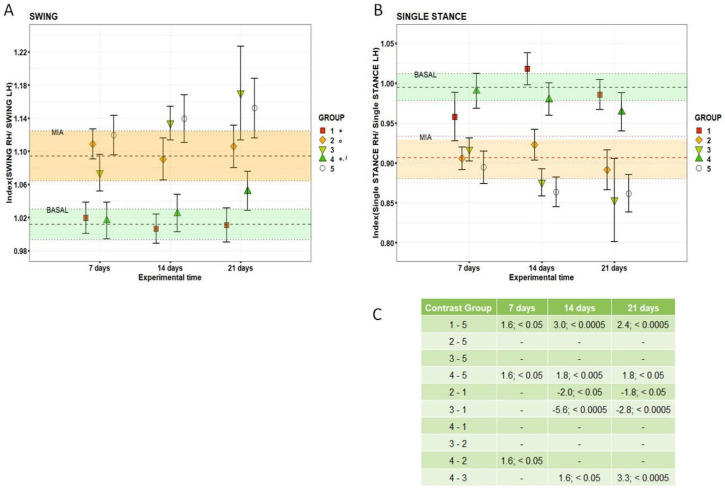
CatWalk results. (**A**) Dot plots of Swing results for each treatment measured at each experimental time. Dots indicate mean and error bars indicate SE. The selected LMM models showed significant effects of treatment (F = 5.35, *p* < 0.005) and experimental (F = 14.43, *p* < 0.0005) time factors. Tables report the pairwise comparisons (*d*-value; *p*-value). Post-hoc pairwise comparisons (1 symbol: *p* < 0.05; 2 symbols: *p* < 0.005; 3 symbols: *p* < 0.0005): * All treatment groups versus NaCl. ° HY-CTL, RV2 and RV4.5 versus TA. # RV2 and RV4.5 versus HY-CTL. § RV4.5 versus RV2. (**B**) Dot plots of Single Stance for each treatment group measured at each experimental time. Single stance values (mean and SE) measured at baseline (BASAL) and seven days after MIA (MIA) are reported as green and orange zones. Dots indicate mean and error bars indicate SE. The selected LMM models showed significant interaction of treatment and experimental time factors (F = 6.71, *p* < 0.0005). (**C**) Single Stance pairwise comparisons (*d*-value; *p*-value). Group 1: TA; group 2: HY-CTL; group 3: RV2; group 4: RV4.5; group 5: NaCl.

**Figure 4 cells-09-01571-f004:**
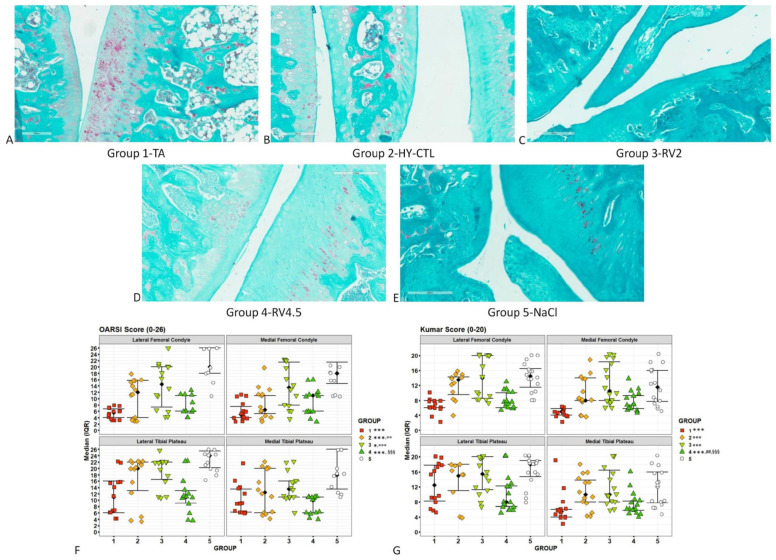
Histopathological assessments of right joints of rats. Histological images of the medial aspect of joints treated with i.a injection of TA (**A**), HY-CTL (**B**), RV2 (**C**), RV4.5 (**D**) and NaCl (**E**). Safranin O/Fast Green staining. Magnification 15×, bar = 200 µm. (**F**–**G**) Scatter plots of Osteoarthritis Research Society International (OARSI) and Kumar results for each treatment group measured at the lateral (femoral condyle and tibial plateau) and medial (femoral condyle and tibial plateau) knee joint compartments; black square dots indicate the median and error bars indicate interquartile range (IQR). Post-hoc pairwise comparisons (1 symbol: *p* < 0.05; 2 symbols: *p* < 0.005; 3 symbols: *p* < 0.0005): * All treatment groups versus NaCl. ° HY-CTL, RV2 and RV4.5 versus TA. # RV2 and RV4.5 versus HY-CTL. § RV4.5 versus RV2. Group 1: TA; group 2: HY-CTL; group 3: RV2; group 4: RV4.5; group 5: NaCl. Black arrowheads: cartilage staining with Safranin O; red arrowheads: fibrillation; FB: fibrous tissue; CL: clones.

**Figure 5 cells-09-01571-f005:**
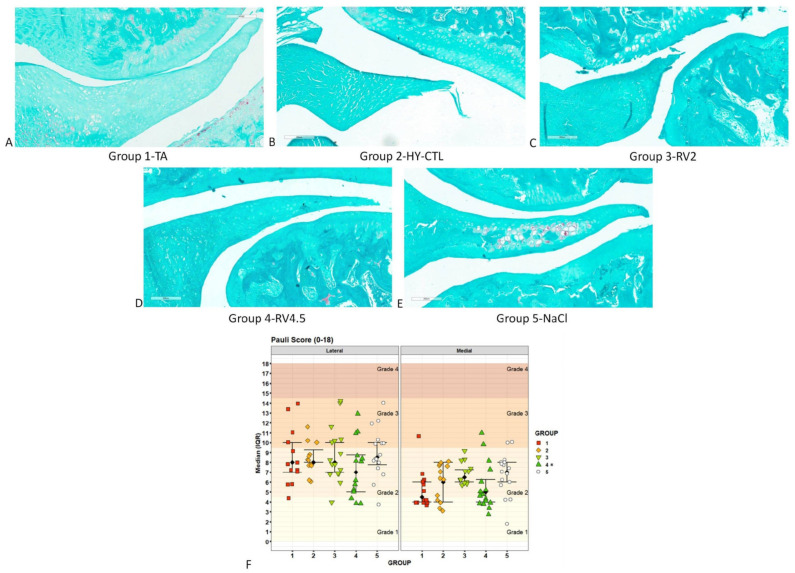
Histopathological assessments of menisci. Histological images of the lateral aspect of menisci treated with i.a injection of TA (**A**), HY-CTL (**B**), RV2 (**C**), RV4.5 (**D**) and NaCl (**E**). Safranin O/Fast Green staining. Magnification 15×, bar = 200 µm. (**F**) Scatter plot of Pauli score results for the histological grading of menisci for each treatment measured at the lateral and medial knee joint compartments; black square dots indicate the median and error bars indicate IQR. The selected cumulative link models (CLM) (likelihood ratio test: χ^2^ = 43.7, *p* < 0.0005) highlighted significant effects of ‘treatment’ (χ^2^ = 13.0, *p* = 0.011) and ‘joint compartment’ (χ^2^ = 31.9, *p* < 0.0005) factors on Pauli score. Post-hoc pairwise comparisons (1 symbol: *p* < 0.05; 2 symbols: *p* < 0.005; 3 symbols: *p* < 0.0005): * All treatment groups versus NaCl. Significantly lower Pauli score results were found for the medial joint compartment than for the lateral one (−29%, *p* < 0.0005). Black arrowheads: undulation; red arrowheads: fibrocartilaginous separation.

**Figure 6 cells-09-01571-f006:**
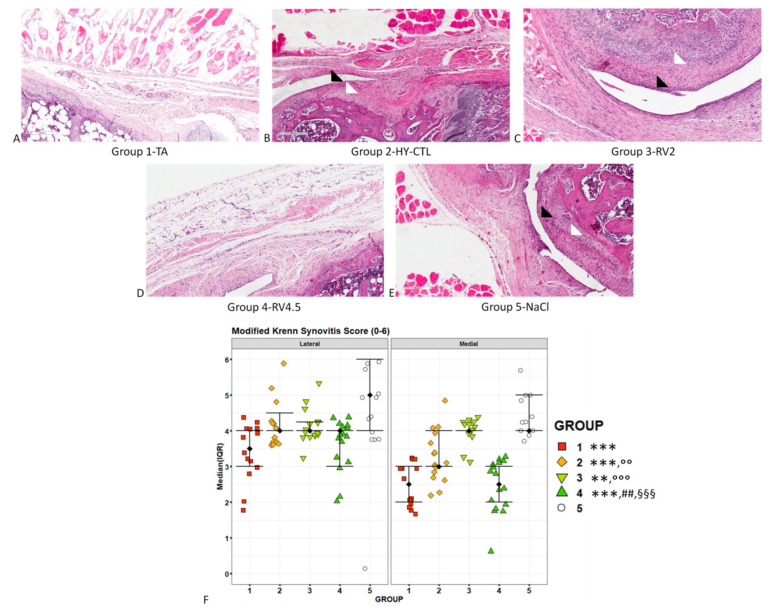
Histopathological assessments of synovial membranes. Histological images of medial synovial membranes treated with i.a injection of TA (**A**), HY-CTL (**B**), RV2 (**C**), RV4.5 (**D**) and NaCl (**E**). Hematoxylin and eosin (H&E) staining, magnification 10×, bar = 200 µm. (**F**) Scatter plot of modified Krenn synovitis score; black square dots indicate the median and error bars indicate IQR. The selected CLM (likelihood ratio test: χ^2^ = 114.6, *p* < 0.0005) takes into account a scale effect of both factors, highlighting their significant effects (‘treatment’: χ^2^ = 79.3, *p* < 0.0005; ‘joint compartment’: χ^2^ = 43.2, *p* < 0.0005) on the modified Krenn synovitis score.Post-hoc pairwise comparisons (1 symbol: *p* < 0.05; 2 symbols: *p* < 0.005; 3 symbols: *p* < 0.0005): * All treatment groups versus NaCl. ° HY-CTL, RV2 and RV4.5 versus TA. # RV2 and RV4.5 versus HY-CTL. § RV4.5 versus RV2. Significant lower modified Krenn synovitis scores were found in the medial joint compartment compared to the lateral one (−15%, *p* < 0.0005). Black arrowheads: enlargement of the synovial lining cell layer; white arrowheads: increased cell density in the synovial stroma.

**Figure 7 cells-09-01571-f007:**
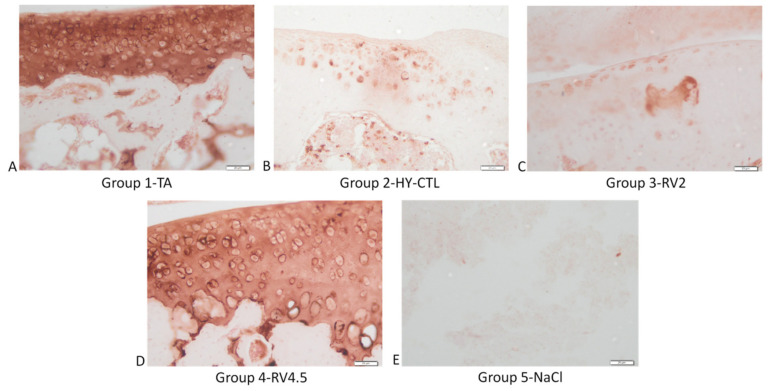
Immunohistochemical assessments of the right knee joints of rats. Images of the medial aspect of the joints treated with i.a injection of TA (**A**), HY-CTL (**B**), RV2 (**C**), RV4.5 (**D**) and NaCl (**E**). Type II collagen staining, magnification 40×, bar = 20 µm.

**Figure 8 cells-09-01571-f008:**
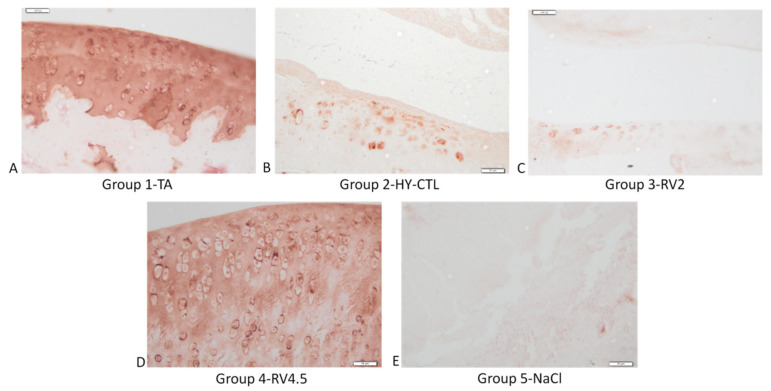
Immunohistochemical assessments of the right knee joints of rats. Images of the medial aspect of the joints treated with i.a injection of TA (**A**), HY-CTL (**B**), RV2 (**C**), RV4.5 (**D**) and NaCl (**E**). Aggrecan staining, magnification 40×, bar = 20 µm.

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
