# Peer review of "Boosting the Intra-Articular Efficacy of Low Dose Corticosteroid through a Biopolymeric Matrix: An In Vivo Model of Osteoarthritis"

_cells, 2020, doi:10.3390/cells9071571_

Round 1
Reviewer 1 Report
This a well written manuscript. Methods are appropriate, results are innovative and supported by a well structured discussion
Author Response
We thank the reviewer for the comments.
Reviewer 2 Report
This manuscript demonstrates the therapeutic effect of a single intra-articular injection of a hyaluronic acid-chitlac (HY-CTL) enriched with triamcinolone acetonide (TA) in a rat model of osteoarthritis. The study was well designed and the concept sounds clinically relevant. Also, the meaningful therapeutic effect of the HY-CTL/TA has been well demonstrated using several different assessment methods. However, this study merely lists the final therapeutic outcome in the animal model and does not go beyond. Although simply showing the final outcome of a new therapeutic modality could be impactful, I think this study is unlikely to gain much interest of readers of the journal Cells. I recommend the authors to submit this study to a more specific journal that aims in treatments of osteoarthritis.
Author Response
We thank the reviewer for the comments.
I and my coauthors sent this manuscript to the journal Cells, specifically to be evaluated for the Special Issue “Hyaluronic Acid: Basic and Clinical Aspects” where I was invited to contribute to; by reading the special issue submission information we thought that our manuscript can be of relevance for this journal, also considering that one of the main application of hyaluronic acid is, currently, for osteoarthritis treatment. Thus, we considered that, on the basis of recent knowledge about the mechanisms that lead to the development of osteoarthritis, the evaluation of a treatment that combine reduction of inflammation and the “stimulation” of cells to repair the damaged tissues, restoring the correct homeostasis, can be of critical importance for this articular pathology. The histopathological analyses presented in this manuscript evaluated, through specific scoring systems, the microanatomy of the osteo-articular tissues but also of the cells that compose this tissue. In addition, according to another reviewer’s suggestion, we now added to the manuscript also the immunohistochemical analyses for type II Collagen and Aggrecan to better evaluate the expression of these main cartilage extracellular matrix components produced by chondrocytes of the articular cartilage, thus to specifically identify antigens (proteins) expression at cellular level (Figure 7 and Figure 8).
Reviewer 3 Report
This is an interesting study evaluating the effects of intra-articular injections of two different concentrations of the corticosteroid triamcinolone (TA) in combination with Hyaluronic Acid (HY)- Chitlac matrix (CTL) in an in vivo chemically induced OA model. The observation in a previous in vitro study of the positive effect of addition of a lactose modified chitosan to HA was experimentally studied in combination with the steroid effect.
The study, well performed and with the results properly reported, lead to the conclusion that treatments with HY-CTL and TA exert beneficial effects in all gain parameters, a reduction of pain, as well as anti-inflammatory and cartilage protective effects.
However, the strength of the results is reduced, not only by the limitations of the study stated by the Authors in the Discussion section, but also because to demonstrate the chondro-protective effect of the proposed treatments, data should be expanded to analyze the expression of the main cartilage ECM components (i.e collagen type II and aggrecan) at gene and/or protein level by means of immunohistochemistry or WB analysis of tissue samples.
Author Response
We thank the reviewer for helpful suggestions aimed at improving our manuscript. After the reviewer suggestion, to better demonstrate the chondro-protective effect of the treatments proposed in the manuscript we analyzed the expression of type II collagen and aggrecan by means of immunohistochemistry in all treatments. These analyses generated new paragraphs in the Materials (2.5 Immunohistochemistry) and Results (3.4 Immunohistochemistry) sections and two new figures (Figure 7 and 8).
Reviewer 4 Report
The manuscript “Boosting the intra-articular efficacy of low dose corticosteroid through a biopolymeric matrix: an in vivo model of osteoarthritis” by Tschon et al is a very interesting study, that take in consideration the effects of intra-articularl administration of Hyaluronic acid a chondroprotective agent, the Chitlac an anabolic stimulator and a anti-inflammatory corticosteroid, in a animal model of OA.
The study is well conducted and well described. To my opinion, just few minor topics should be addressed.
In the histological analyses in figures 4, 5 and 6, arrows should be added to highlight the results, in this way they become clearer even for people who are not expert in histochemical analyses.
The lateral legend of OARSI, Kumar and Pauli scores in Figures 4, 5 and 6 should be enlarged to be more readable.
Author Response
We thank the reviewer for the comments and we now added arrows and enlarged the lateral legends of OARSI, Kumar and Pauli scores in the Figures 4, 5 and 6.
Moreover, according to another reviewer’s suggestion, we performed immunohistochemical analyses for type II Collagen and Aggrecan for demonstrating the chondro-protective effect of the treatments. These analyses generated new paragraphs in the Materials and Results sections and two new figures (Figure 7 and 8).
Round 2
Reviewer 2 Report
I feel the current manuscript meets the standards for the journal Cells and suggest to proceed further for publication.
Reviewer 3 Report
I have no further questions for the authors